# The Current State of the Diagnoses and Treatments for Clear Cell Renal Cell Carcinoma

**DOI:** 10.3390/cancers16234034

**Published:** 2024-12-01

**Authors:** Anthony E. Quinn, Scott D. Bell, Austin J. Marrah, Mark R. Wakefield, Yujiang Fang

**Affiliations:** 1Department of Microbiology, Immunology & Pathology, Des Moines University, West Des Moines, IA 50266, USA; anthony.e.quinn@dmu.edu (A.E.Q.); scott.d.bell@dmu.edu (S.D.B.); 2Department of Surgery, University of Missouri School of Medicine, Columbia, MO 65212, USA; ajm4cc@missouri.edu (A.J.M.); wakefieldmr@health.missouri.edu (M.R.W.); 3Ellis Fischel Cancer Center, University of Missouri School of Medicine, Columbia, MO 65212, USA

**Keywords:** clear cell renal cell carcinoma, kidney cancer treatment, kidney cancer epidemiology

## Abstract

Clear cell renal cell carcinoma is a kidney cancer with high incidence rates in the western world. There is an increasing research base fueling the modern application of therapy to treat this cancer. Despite the traditional surgical interventions available for treatment, more is now understood about its metastases and optimal treatment strategies. The treatment of clear cell renal cell carcinoma is quickly advancing to ensure that patients receive personalized cancer therapy specific to their solid tumor. While surgical interventions remain the dominant treatment modality, adjuvant drug therapies have shown promising results, especially for patients with advanced disease.

## 1. Introduction

Kidney cancer accounts for about 2% of all cancer diagnoses, having the highest prevalence in men, with a relative risk (RR) of 1.7 for men compared to women [1]. For the past three decades, the rates of renal cell carcinoma (RCC) have been rising in the United States and Europe [2], accelerating the need for new and improved therapies. Cancer originating from the renal epithelium, termed RCC, is the most common form of kidney cancer, accounting for 80% of malignant kidney tumors [3]. RCC is further subdivided into tumor pathological classifications: clear cell, chromophobe, and papillary tumors [4]. Clear and papillary tumors originate from the proximal tubular cells, while chromophobe tumors originate from intercalated cells [5]. Clear cell renal carcinoma (ccRCC) accounts for 75% of kidney tumor diagnoses. It also has the greatest tendency to metastasize, leading to a low 5-year survival rate of 44–69% as opposed to 82–92% for papillary tumors and 78–92% for chromophobe tumors [6,7].

Clear cell tumors exhibit hematogenous spread [8]. Additionally, these tumors often extend directly into major blood vessels near the kidneys, specifically the renal veins and the inferior vena cava [8]. This direct extension into major vessels facilitates metastasis. The major sites of dissemination are the lungs, liver, lymph nodes, brain, bone, and adrenal glands [9].

One of the defining histopathological characteristics of ccRCC is the clear soap bubble appearance of the tumor cells, visibly seen under a microscope when dissolving lipid droplets within the cell with alcohol during a hematoxylin and eosin stain [10]. Some cells may present with eosinophilic granular cytoplasms [5]. Necrosis is more frequently observed in larger renal lesions, typically seen in those exceeding 4 cm in size. The incidence and extent of necrosis are also linked to high-grade tumor histology [11].

One of the most notable genetic disorders associated with clear cell renal cell carcinoma (ccRCC) is Von Hippel–Lindau (VHL) disease, an autosomal dominantly inherited disorder [12]. The VHL gene, which is located on the short arm of chromosome 3, functions as a tumor suppressor gene [12]. Patients with VHL disease have an inactivated germline mutation in one allele of the VHL gene. This germline mutation in the VHL gene increases the risk of developing highly vascularized tumors in organs like the kidney, adrenal gland, and pancreas [13]. Around 45% of all people diagnosed with VHL disease develop RCC [14]. Furthermore, the inactivation of the VHL gene is the most common alteration in renal carcinomas, occurring in 50–70% of sporadic cases [15,16]. RCC is also associated with tuberous sclerosis complex (TSC). TSC is an autosomal-dominant disorder caused by mutations in the TSC1 or TSC2 genes, leading to the formation of hamartomas in multiple organ systems [17]. About 2–4% of patients with TSC develop RCC [18].

Paraneoplastic syndromes (PNSs) are seen in up to 40% of patients with RCC [19]. The PNS symptoms associated with RCC vary from constitutional symptoms like fever, cachexia, and weight loss to metabolic abnormalities like hypercalcemia [20]. PNSs are unrelated to the local or distant spread of a tumor and hence are not indicative of metastatic potential [20]. It is theorized that PNSs arise from tumors producing biologically active substances (i.e., hormones), the modulation of the immune system, immune complex production, and immune suppression [21]. The presence of PNS symptoms can be an important diagnostic tool as the commonly associated symptoms are usually not present until the later stages of the disease.

The rising prevalence of kidney cancer can be attributed to a troublesome combination of factors, some of which arise from increasingly concerning modifiable risk factors. Smoking, hypertension, and obesity are all risk factors for RCC but require patients to make often challenging lifestyle changes [22].

One of the major modifiable risk factors in the pathogenesis of RCC is smoking. Hunt et al. performed a meta-analysis of 24 studies conducted in Europe, North America, and Australia. They found that the RR of smokers developing RCC compared to nonsmokers was ~1.38 for both men and women [23]. The study also found that smoking duration and the number of cigarettes smoked was directly related to RCC risk [23]. In a study consisting of ~250,000 United States veterans over the age of 26, the relative risk factor for RCC grew from 1.37 for < 10 cigarettes per day up to 2.06 for ≥40 cigarettes per day [24]. In comparison to nonsmokers, smokers with RCC were more likely to have concurrent diseases including coronary artery disease and chronic obstructive pulmonary disease, which potentially account for the overall lower survival rate in smokers [25].

Obesity, defined as a body mass index (BMI) of ≥30 kg/m^2^ by the World Health Organization, has been shown to lead to an increased risk of RCC in both men and women [26]. The prevalence of obesity in the United States is ever-growing. In 2017–2018, the number of American adults who were considered obese was 42.8% or 76 million people [27]. The number of children classified as obese in the United States jumped from 14.7% in 1999 to 19.2% in 2018 [28]. As one in five children and adolescents are considered obese in the U.S., it is thought that the incidence of RCC will increase alongside the rise in obesity. In a quantitative review of 14 studies by Bergström et al., it was estimated that the RR of RCC for each unit increase in BMI was 1.07 [26]. Furthermore, obesity can lead to tissue hypoxia and renal damage, leaving patients more susceptible to tumor formation [29]. Additionally, RCC has been shown to elevate insulin and insulin-like growth factor-1 receptor proteins, leading researchers to believe obesity plays a key role in RCC progression [30].

Hypertension has also been shown to be a significant risk factor for RCC. In a study from 2013, over 249 cases of RCC were analyzed for significant risk factors. The hazard ratio between hypertension and RCC was found to be 1.7, meaning that individuals with hypertension had a 70% higher risk of developing RCC compared to individuals without hypertension [31]. Furthermore, a prospective study that followed 36,728 women and 35,688 men over 18 years of medical follow-ups found that women with systolic blood pressure levels of 130–149 mm Hg, 150–169 mm Hg, and ≥170 mm Hg, compared to those with levels < 130 mm Hg, had relative risks of renal cell cancer of 1.7, 2.0, and 2.0, respectively [32]. While the exact association between RCC and hypertension remains unknown, it is theorized that renal hypoxia arising from hypertension contributes to tumor progression through the transcription factor known as hypoxia-inducible factor [33]. Additionally, hypertension has shown increased rates of lipid peroxidation, which has can promote renal tumor progression [34].

ccRCC exhibits a complicated disease manifestation that can make early-stage diagnosis difficult [35]. Although early detection is known to be crucial for achieving the best outcomes, many patients are diagnosed with advanced disease. For instance, in England in 2017, the data revealed that among patients with a documented stage at diagnosis, 19% were at stage III and 23% were at stage IV when they first presented [36]. The traditional triad of hematuria, flank pain, and abdominal mass is now considered rare. Symptoms, if they appear, are often vague, nonspecific, and delayed in onset [36], shifting care from preventative to palliative.

To only further underscore the complex nature of treating RCC, the previously used treatments relied on cytokine-based drug therapies and showed poor response rates of only 10–15% [37]. These therapies also had considerable side effects, including neuronal and cardiovascular toxicities [38]. Moreover, RCC characteristically shows resistance to chemotherapy and radiation therapies [39]. The use of such toxic and relatively ineffective therapies, alongside the late presentation of the disease and the rising prevalence of risk factors, there is an increasing need for effective and timely treatments.

With the rate at which RCC cases are rising, there has been a call for a wider array of therapeutic options and refined treatment methods. The standard treatment option for RCC is nephrectomy. Radical nephrectomy is typically reserved for tumors greater than T1 in staging course, and partial nephrectomy is typically used for tumors that are at T1 staging. RCC is significantly resistant to chemotherapeutic treatment options; thus, the amounts of neoadjuvant and adjuvant chemotherapy treatments have steadily become less utilized for RCC tumors. Researchers have begun to investigate new classes of immunotherapeutic drugs capable of targeting specific points in immunoregulation and destruction of RCC and ccRCC tumor cells.

## 2. Diagnoses

### 2.1. Clinical Presentation

The early stages of RCC are generally asymptomatic, making diagnosis difficult (Figure 1). The distinctive triad of clinical symptoms found in RCC are flank pain, hematuria, and a palpable abdominal mass, which are not apparent until advanced disease progression [40]. Extremity edema and right-sided varicocele due to testicular venous obstruction and tumor occlusion of the inferior vena cava, respectively, are also symptoms that most often present in the later stages of RCC [40,41]. As the symptomatic presentation of RCC usually occurs in advanced stages, it is often diagnosed incidentally. It is estimated that less than 30% of cases of RCC are diagnosed based on clinical symptoms alone [41]. To further underscore the absence of symptoms in RCC, it is estimated that greater than 50% of RCCs are detected incidentally [41].

### 2.2. Laboratory Testing

Gross hematuria, a common manifestation of RCC, warrants diagnostic tests like microscopic urinalysis and management per the recommendations of the American Urological Association [24]. Additional laboratory testing that should be conducted in a workup of RCC includes blood levels of creatinine, lactate dehydrogenase (LDH), and C-reactive protein (CRP), serum calcium, as well as a complete blood count (CBC). These tests are often used to evaluate for polycythemia and hypercalcemia, which are common presentations in PNSs [34].

### 2.3. Imaging

Ultrasonography (US) is usually the first imaging modality conducted when there is a presumptive renal mass and allows for additional assessments like renal function and vascularity. US is also instrumental in differentiating cysts from hypovascular solid tumors observed on CT scans and is more effective at revealing septations in cases of complex cystic lesions [42].

Contrast-enhanced computed tomography (CT) is the current standard for diagnosing and staging RCC, with a nearly 91% accuracy rate [43,44]. CT scanning of the thorax, abdomen, and pelvis is considered mandatory for accurate diagnosis. Magnetic resonance imaging (MRI) of the brain and a bone scan are also typically included in an RCC workup, alongside US and CT, when tumor metastasis is suspected [45]. A noncontrast CT with an MRI of the abdomen is preferred in patients with suspected RCC with renal insufficiency, pregnancy, or an allergy to intravenous contrast [43].

The use of fluorodeoxyglucose-positron emission tomography (FDG-PET) is often limited in RCC cases due to its false negative results caused by normal physiological excretion by the kidneys. However, ccRCC has been seen to have a much greater tumor-to-nontissue uptake, as well as standardized uptake in 18FDG-PET scanning [35].

ccRCC manifests with specific imaging features. ccRCC tends to grow outward from the kidney and appears varied in texture due to the heterogeneity of the cells within the tumor or bleeding [46]. It is also typically a highly vascularized tumor [9]. ccRCC also shows noticeably more intense contrast enhancement on CT when compared to other RCC subtypes [47].

### 2.4. Biopsy

Solid tumors are managed based on size, with masses less than 1 cm typically being observed and those greater than 1 cm usually excised or biopsied [40]. The role of renal mass biopsy in diagnosis and treatment has somewhat increased as active surveillance has become a viable option for RCC. However, biopsies carry an increased risk of false negative results and concerns remain about the nondiagnostic rate and negative predictive value [40].

Because of this, biopsies of renal lesions are generally not performed unless the results would impact therapy recommendations. This would include patients with multiple tumors or specific underlying conditions [48]. Renal biopsies may be considered in patients with RCC who have upcoming ablative or thermal therapy to determine the histologic subtype of the tumor [48].

While the diagnostic accuracy for biopsies of RCC is generally high with a low complication rate, renal biopsy should be used selectively, and further investigation is needed to determine its clinical application [49].

### 2.5. Staging

RCC has transitioned from the Robson system to the Tumor, Node, and Metastasis (TNM) system [50]. Traditionally, the Robson system corresponded with surgical staging and referenced the tumor’s association within the renal capsule, perirenal fat penetration, renal vein invasion, and lymph node metastases [51]. The TNM system follows a similar approach but accounts for the size, perinephric fat invasion, and interior vena cava wall invasion [50].

The 8th edition of The American Joint Committee on Cancer (AJCC) has guidelines to formally stage RCC kidney tumors (Table 1). Tx defines a primary tumor that cannot be assessed, and T0 shows no evidence of a primary tumor [52]. T1 represents a tumor ≤ 7 cm limited to the kidney. T1 is subdivided into T1a for a tumor ≤ 4 cm and T1b for a tumor > 4 cm but ≤7 cm [52]. T2 represents a tumor > 7 cm limited to the kidney. T2 is subdivided into T2a for a tumor > 7 cm but ≤10 cm and T2b for a tumor > 10 cm [52]. T3 represents a tumor that extends into the major veins or perinephric tissues but does not extend into the ipsilateral adrenal gland or Gerota’s fascia. T3 is subdivided into separate classifications, T3a is used when a tumor extends into the renal vein, its branches, or invades into the pelvicalyceal system, perineal or renal sinus fat. T3b describes a tumor extending into the vena cava underneath the diaphragm. T3c describes a tumor extending into the vena cava above the diaphragm or invading the vena cava wall [52]. T4 represents a tumor invading beyond Gerota’s fascia, including extension to the ipsilateral adrenal gland [52].

Nx defines a regional lymph node that cannot be assessed, and N0 shows no evidence of lymph node metastasis [52]. N1 represents metastasis to the regional lymph nodes, and N2 is not defined according to the current and proposed 8th edition AJCC RCC tumor staging. M0 defines no metastasis, and M1 defines distant metastasis [52].

Following the AJCC guidelines, T1 and T2 tumors are based on size alone, while T3 and T4 represent the extension of the tumor beyond the kidney. Overall, the TMN system allows for an accurate diagnosis by describing more pathological features.

## 3. Treatment

### 3.1. Radical Nephrectomy

Radical nephrectomy (RN) of the kidney is the standard in the removal of large RCC and ccRCC tumors when the contralateral kidney is fully functional [53]. RN typically involves the removal of the pathologic kidney and Gerota’s fascia. When removing Gerota’s fascia during RN, the adrenal gland is commonly removed too unless tumorigenic spread to the adrenal gland is not found. Currently, RN is commonly performed in open, laparoscopic, or robot assisted settings. Both open radical nephrectomy (ORN) and laparoscopic radical nephrectomy (LRN) have similar rates of complications and mortality during surgery and have similar oncologic effectiveness. However, LRN is advantaged as patients experience less postoperative pain, a faster return to activity, and shorter hospitalization time [54,55]. However, ORN is often the surgical approach taken in more complex cases of RCC as it allows for a wider array of surgical tools and techniques to be used [56].

Like many surgical procedures today, RN is advancing quickly using a growing body of robot-assisted surgical techniques and research. Compared to ORN, robot-assisted radical nephrectomy (RARN) results in fewer surgical complications and decreased hospitalization times [57]. LRN and RARN have been shown to have similar surgical outcomes, adverse event rates, and lengths of hospitalization, with the caveat being that RARN is currently more expensive for hospital systems to support [56,57]. RARN is being adopted at a slower pace in hospitals, but, as the technology becomes more accessible and the advantages become more pronounced, more systems are likely to utilize and engage with the technology.

### 3.2. Partial Nephrectomy

Partial nephrectomy (PN), also known as nephron-sparing surgery (NSS), is considered the standard for small renal masses and is increasing in utilization in modern hospital settings [58,59]. PN is typically utilized for RCC tumors found near T1 staging [60]. PN involves the removal of unhealthy renal tissues with tumor involvement and leaves patients with healthy, functional renal tissues intact. Like RN, PN can be performed in open and laparoscopic settings. Laparoscopic partial nephrectomy (LPN) has been found to require less ischemia time, with similar functional and oncologic outcomes to open partial nephrectomy (OPN) [61]. A drawback of LPN is that the operation is technically challenging, and surgeons need to be trained on the correct procedure to perform the surgery successfully.

Kunath et al. found that compared to RN, PN shows similar surgery-related mortality rates and cancer-specific survival times when performed to remove RCC [62]. Similarly, Dash et al. found no evidence of a poorer cancer prognosis when comparing PN and RN to remove ccRCC tumors of 4–7 cm [63]. Thompson et al. found that patients who had received RN or PN had similar overall survival, but, interestingly, the patients that received RN were more likely to die from RCC later on [64]. Compared to RN, PN is associated with a greater health-related quality of life after surgery for renal tumors [63]. Recently, in a meta-analysis, Yang found that PN was associated with fewer incidences of hospital mortality and lower reoperation rates for patients with RCC [65].

More recently, robotic-assisted partial nephrectomy (RAPN) has become a tool in the surgeon’s arsenal to reduce RCC tumors. RAPN has been indicated as a safe approach for complex renal lesions that score high on the RENAL nephrectomy scoring system [66]. These high-scoring renal lesions would historically be treated with OPN or RN. Robot assistance has improved LPN by decreasing the learning curve associated with LPN, increasing the utility of surgical tools, and standardizing the LPN procedure [67]. Like RARN, RAPN is more expensive compared to its laparoscopic counterpart, but efficient hospital systems and the saturation of robot-assisted technologies will begin to decrease this cost difference [68]. Recently, it has been shown that when performing either RAPN or LPN, the surgeon’s experience level does not play a role in the functional recovery of the kidneys after nephrectomy, highlighting the idea that further research is needed to uncover what aspects of RAPN and LPN are most beneficial for positive functional outcomes [69].

Further optimization of PN such as the addition of hemostatic agents, like TachoSil and FloSeal, after renorrhaphy has been attempted. However, the use of these hemostatic agents did not confer any benefit after PN [70]. Recently, research has shown that consideration needs to be emphasized when performing PN or RN of patients who have had heart valve replacement operations [71]. It was found that in patients that had received an artificial heart valve, there was an increase in cardiac complications, intraoperative complications, and longer hospital stay times after PN surgery [71]. Furthermore, it was found that in patients that had received an artificial heart valve, there was an increase in cardiac complications, postoperative bleeding, and intraoperative complications after RN surgery [71]. Importantly, it was found that race/ethnicity matters when considering treatment with RN or PN for ccRCC stage T1aN0M0 patients [72]. When compared to Social Security Administration life expectancy tables, African American patients showed shorter life expectancies after PN or RN than what was shown in the Security Administration life expectancy tables [72], but Caucasians, Hispanics/Latinos, and Asians showed life expectancies that were consistent with the tables after PN or RN.

Preoperative nephrometry scoring systems are tools used in assessing the complexity of renal tumors and predicting the potential complications associated with partial nephrectomy. The Preoperative Aspects and Dimensions Used for an Anatomical (PADUA), RENAL nephrectomy scores, and Simplified PADUA Renal (SPARE) nephrometry score are among the most utilized systems.

The PADUA score, developed in 2009 by Ficarra et al., evaluates the clinical tumor size along with the anatomical features related to anterior or posterior tumor location, pole positioning, rim positioning, exophytic or endophytic growth, and involvement of adjacent structures of either the renal sinus, collecting system, or both. The total calculated score, based on the sum of the aforementioned factors, is considered predictive of postoperative outcomes following partial nephrectomy [73,74]. A systematic review and meta-analysis compared the PADUA, RENAL, and C-index scores in predicting outcomes after partial nephrectomy. The PADUA score demonstrated a significant correlation with warm ischemia time, estimated blood loss, and operation time, emphasizing its utility in distinguishing surgical complexity [75].

The RENAL nephrometry score (RNS), introduced in 2009 by urologists Alexander Kutikov and Robert G Uzzo, assesses tumors based on radius (tumor size), exophytic/endophytic properties, nearness to the collecting system, anterior/posterior location, and location relative to polar lines [76]. Studies have demonstrated that a higher RNS correlates with longer ischemia times and a greater likelihood of postoperative complications. A prospective study of 150 renal masses treated with laparoscopic NSS evaluated the RNS in predicting adverse outcomes. Patients were categorized into low-risk (RNS ≤ 6) and high-risk (RnS ≥ 7) groups. It was found that the high-risk patient group had significantly higher rates of adverse outcomes compared to the low-risk group, with 44.26% and 25.84%, respectively [77]. Similarly, in a prospective study of 20 patients undergoing open partial nephrectomy, a higher RNS was associated with increased blood loss, longer warm ischemia times, greater declines in glomerular filtration rate, and higher complication rates [78]. Tumor complexity and risk factors for progressive renal disease were significant predictors of these outcomes, highlighting the utility of RNS in nephrectomy evaluation.

One of the newer nephrometry scoring systems that has been refined in the last five years is the SPARE nephrometry system. It is a more streamlined version of the PADUA score, incorporating only four of the previous system’s variables. These include rim location, renal sinus involvement, exophytic rate, and tumor size [79]. Research indicates that the SPARE score maintains predictive accuracy for postoperative complications comparable to the original PADUA score, offering a more straightforward assessment method. In a multi-institutional study involving 536 patients undergoing RAPN, the SPARE system demonstrated a higher predictive accuracy, with an area under the curve (AUC) for surgical success of 0.73 compared to PADUA (AUC: 0.65) and RENAL (AUC: 0.68) [80]. Additionally, an external validation study involving 202 RAPN cases in a single center found that the SPARE scores were comparable to PADUA and RENAL for predicting complications, with area under the curve values of 0.61, 0.59, and 0.57, respectively, and a *p*-value of 0.43 [81].These findings highlight SPARE as a simple and reliable tool for assessing risks during nephrometry evaluations, while offering equal or even superior accuracy in predicting outcomes.

Recent research has investigated the impact of imaging modalities on the application of nephrometry scoring systems in clinical practice. A study evaluating the impact of preoperative imaging on the accuracy and consistency of RENAL and PADUA scoring in predicting surgical outcomes for NSS found that including the excretory phase in imaging improved consistency among urologists in scoring tumor complexity but did not significantly enhance predictions of surgical outcomes like warm ischemia time. The study also found that both CT and MRI work equally well for scoring and that a two-plane dataset without excretory phase is sufficient for routine clinical use [82]. These findings support simplifying imaging protocols while maintaining the accuracy of PADUA and RENAL scores for planning partial nephrectomy.

Contemporary studies have also explored the relationship between nephrometry scoring surgical experience and functional outcomes. In a multi-institutional analysis of over 4000 patients aged 30–80 with two kidneys, PADUA 6–9 renal masses, and a preoperative eGFR > 45 mL/min/1.73 m^2^, the outcomes of laparoscopic or robot-assisted partial nephrectomy were evaluated. It was found that perioperative outcomes such as ischemia time and acute kidney injury were more strongly associated with nephrometry scores and patient factors than surgical experience. Functional recovery, including retaining at least 90% of baseline eGFR one year postoperatively, was not significantly influenced by the surgeon’s experience [69]. These findings reinforce the value of nephrometry scores in providing a standardized approach to tumor complexity assessment, which can guide surgical planning regardless of the surgeon’s experience.

### 3.3. Chemotherapy

Due to the chemotherapy-resistant nature of RCC, chemotherapeutic agents are typically not the standard treatment used for RCC or ccRCC unless the cancer has been unresponsive to other therapies [83]. The area of chemotherapeutics in the treatment of RCC and ccRCC is still a growing field that is being refined with new drugs, dosage strategies, and combination therapies. Many non-randomized phase I and phase II clinical trials have yielded less than satisfying results in the treatment of RCC when compared to other therapeutic strategies [84]. However, it has been shown that chemotherapy can be more effective in cases of RCC and ccRCC with sarcomatoid differentiation [84,85]. Nanus et al. showed that combination chemotherapy with doxorubicin and gemcitabine gave good oncologic results in patients with RCC with sarcomatoid differentiation [85].

### 3.4. Tyrosine Kinase Inhibitors

There are increasing numbers of targeted therapeutic options being utilized for the treatment of RCC and ccRCC due to the chemotherapy-resistant nature of the cancers.

In 2006, sunitinib was approved by the FDA for the treatment of RCC. Sunitinib is a receptor tyrosine kinase inhibitor (TKI) that can block the action of receptors for vascular endothelial growth factor (VEGF), which can aid in RCC tumor metastases via enhancing angiogenesis (Figure 2) [86,87]. However, sunitinib, as a first-line treatment option, has been replaced by combination TKI and immune checkpoint inhibitor (ICI) therapies as it was found to be inferior in multiple studies [88,89,90,91].

Cabozantinib, another TKI, inhibits a wide range of tyrosine kinase receptors, including ET, VEGFR-1-3, KIT, TRKB, FLT-3, and TIE-2. Notably, unlike other TKIs, it also demonstrates significant activity against MET and AXL [88]. In multiple studies comparing cabozantinib to sunitinib, cabozantinib outperformed sunitinib in progression-free survival (PFS) [92,93]. In the CheckMate 9ER trial, a combination of nivolumab at 240 milligrams biweekly and cabozantinib at 40 milligrams daily was compared to sunitinib monotherapy at 50 milligrams daily for four weeks in a six-week cycle in patients with previously untreated advanced clear cell RCC. The ICI and TKI combination demonstrated superior outcomes, with a median PFS of 16.6 months compared to 8.3 months and an overall survival (OS) benefit. Patients in the combination group also reported a higher health-related quality of life [89]. Now, the guidelines for metastatic RCC recommend 40 mg of cabozantinib daily in combination with nivolumab as first-line treatment and as monotherapy at 60 mg daily as second-line treatment [88].

Like sunitinib and cabozantinib, the TKI axitinib also inhibits VEGFR-1-3 and, when in combination with ICI, demonstrates powerful therapeutic effects. The KEYNOTE-426 phase 3 study revealed the superior efficacy of pembrolizumab and axitinib compared to sunitinib for the first-line treatment of advanced ccRCC. Patients treated with pembrolizumab at a dose of 200 mg administered intravenously every three weeks and axitinib at 5 mg taken orally twice daily showed better outcomes in OS, PFS, and objective response rate compared to those treated with sunitinib at 50 mg taken orally once daily in a four-week cycle [94]. The combination of pembrolizumab and axitinib is now recommended as a first-line treatment of metastatic RCC based on the results of the study [88].

Minardi et al. found that increased VEGF levels in metastatic RCC tumors directly correlated to shorter patient survival [87]. Furthermore, ccRCC has been shown to have increased levels of VEGF compared to other RCC subtypes. Thus, the targeting of VEGF in the treatment of ccRCC will likely be a continually growing area of research, especially in combination with ICI inhibitors [95].

### 3.5. mTOR Inhibitors

The mTOR pathway serves as a central integrating signaling pathway for the growth, spread, and survival of cells [96]. mTOR signaling is upregulated past normal levels in nearly 70% of all cancers. Thus, the pathway serves as a viable target for cancer therapy and is receiving an increased amount of interest in pharmacologic cancer treatment research [97]. Robb et al. found that mTOR signaling is specifically upregulated in ccRCC cell lines and that treatment with a rapamycin regimen inhibited the proliferation of the ccRCC cells [98]. In a phase III trial for patients who had tumor progression after treatment with VEGF-targeted therapies, everolimus, an mTOR inhibitor, was found to increase progression-free survival compared to placebo [99]. Furthermore, in a phase II study, patients were treated with an everolimus regimen, and progression-free survival was found to be greater than or equal to 6 months in nearly 70% of those treated [100]. In 2007, the FDA approved temsirolimus for the treatment of advanced RCC. Temsirolimus is another mTOR inhibitor that has shown enhanced progression-free survival in patients that have received prior immunotherapies [101]. Furthermore, temsirolimus has been indicated for use in patients who present with RCC and multiple factors predictive of short survival [102].

### 3.6. Immune Checkpoint Inhibitors

Immune checkpoint inhibitors are drugs that target specific biological processes in the immune response. In the treatment of cancer, there is an increasing body of work examining the immune checkpoint inhibition of programmed death 1 (PD-1) and its ligand, programmed death ligand 1 (PD-L1). The PD-1/PD-L1 axis controls the activation of T cells. Without inhibition, these T cells mediate the destruction of cancer cells, but cancer cells are able to control this axis to cause the inhibition of T-cell targeting and prevent destruction. In ccRCC, increased levels of PD-1 and PD-L1 have been shown to correlate with more adverse clinicopathological features and poorer outcomes [103]. In the phase IIIb/IV CheckMate 374 study, the efficacy of the administration of 240 mg of nivolumab, a monoclonal antibody to PD-1, every two weeks for patients who had already been treated for ccRCC was validated (Figure 2) [104]. Furthermore, when compared to everolimus, nivolumab showed increased overall survival for patients that had been previously treated for advanced RCC [105]. ICIs in immunotherapy in general have the potential to become the standard of care for the treatment of advanced ccRCC [106]. The use of these immunotherapeutics is being heavily explored in the adjuvant setting, following nephrectomy.

## 4. Future Perspectives

Unfortunately, nearly 35% of RCC recurs after gold-standard nephrectomy treatment [107,108]. Recently, there been a major push in ccRCC treatment to optimize adjuvant therapy to decrease postsurgical ccRCC recurrence. Though EGFR tyrosine kinase inhibitors show efficacy as neoadjuvant therapies, it has been shown that they are less effective after nephrectomy [109,110]. Also, EGFR tyrosine kinase inhibitors have considerable toxicity in adjuvant treatments, thus causing a large percentage of patients to stop their adjuvant usage [111]. Pembrolizumab, an ICI that targets the CD8+ T-cell PD-1 receptor similarly to nivolumab, is showing increased efficacy in adjuvant applications after nephrectomy [109,112]. The KEYNOTE-564 trial showed that using pembrolizumab as an adjuvant monotherapy, when compared to a placebo, resulted in superior disease-free survival [113]. To monitor and modulate the progression of ccRCC, novel research is studying the use of miRNAs. Recently, several miRNAs have shown the ability to track the disease progression of ccRCC, thus adding another tool to the arsenal of biomarkers to track cancer progression during various phases of treatment [114].

## 5. Conclusions

ccRCC is becoming an increasing concern in health systems and it needs to be recognized in its earlier stages to avoid adverse consequences. Due to the lack of RCC screening, many patients rely on incidental diagnosis when receiving imaging for unrelated issues. RCC screening can be implemented with simple laboratory tests at a low cost to the hospitals to rule out concern and avoid these incidental, late-stage findings. While RN and PN remain the predominant treatment modalities for ccRCC, new targeted therapies are being developed that are noninvasive with promising results.

## Figures and Tables

**Figure 1 cancers-16-04034-f001:**
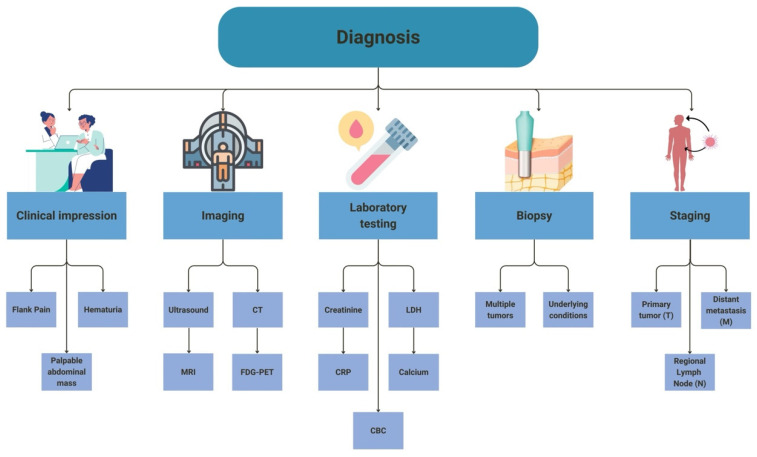
Diagnostic features and modalities for ccRCC. (CT—computed tomography, MRI—magnetic resonance imaging, FDG-PET—fluorodeoxyglucose-positron emission tomography, CRP—C-reactive protein, LDH—lactate dehydrogenase, CBC—complete blood count, ccRCC—clear cell renal cell carcinoma).

**Figure 2 cancers-16-04034-f002:**
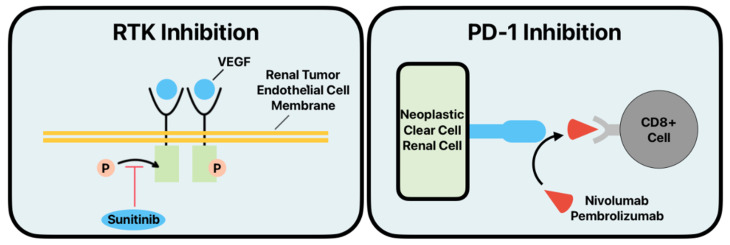
Mechanism of action of the immunotherapeutic agents sunitinib and nivolumab on neoplastic clear cell renal cell carcinoma cells. (RTK—receptor tyrosine kinase, PD-1—programmed death 1 protein, VEGF—vascular endothelial growth factor).

**Table 1 cancers-16-04034-t001:** RCC staging following AJCC guidelines.

**Primary Tumor (T)**	
Tx	Primary tumor not assessed
T0	No evidence of primary tumor
T1	Tumor ≤ 7 cm, limited to kidney
T1a	Tumor ≤ 4 cm, limited to kidney
T1b	Tumor > 4 but ≤7 cm, limited to kidney
T2	Tumor ≤ 7, limited to kidney
T2a	Tumor > 7 cm but ≤10 cm, limited to kidney
T2b	Tumor > 10 cm, limited to kidney
T3	Tumor extends into major veins, perinephric tissue but not ipsilateral adrenal gland or Gerota’s fascia
T3a	Tumor extends into renal vein, its branches, or pelvicalyceal system, perineal or renal sinus fat
T3b	Tumor extends into vena cava underneath diaphragm
T3c	Tumor extends into vena cava above diaphragm or vena cava wall
T4	Tumor invades past Gerota’s fascia
**Regional Lymph Node (N)**	
Nx	Regional lymph node not assessed
N0	No evidence of regional lymph node metastasis
N1	Regional lymph node metastasis
N2	Not defined
**Distant metastasis (M)**	
M0	No distant metastasis
M1	Distant metastasis

## Data Availability

No new data were created or analyzed in this study. Data sharing is not applicable to this article.

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
