# Peer review of "The Current State of the Diagnoses and Treatments for Clear Cell Renal Cell Carcinoma"

_cancers, 2024, doi:10.3390/cancers16234034_

Round 1

Reviewer 1 Report

Comments and Suggestions for Authors

The current review provides an overview of clear cell renal cancer. The authors valuablely address the epidemiology, diagnosis, and treatment. It sounds more like a commentary than a systematic or narrative review. It should be improved with a "Materials and Methods" paragraph that may highlight the search keyword and the number of papers on the field. The paper lacks several items, such as:

- Preoperative scores predicting complications within the Partial nephrectomy part (PMID 39408076, 39369793, 34528773, 29715137) . PADUA, RENAL, SAFE as well as SPARE scores are commonly used in renal practice and should be added in a separate paragraph;

- "Future Perspective" paragraph;

- papers on positive margin status and further treatments after PN (PMID 25450033, 29715137), sarcomatoid variants, the use of hemostatic agents (PMID 25139104).

Author Response

Thank you for reviewing our above-named manuscript submitted to MPDI Cancers.  We have made changes requested by the reviewers in our revision or addressed the concerns in this response, as detailed below and/or shown in the revised manuscript.

Reviewer #1: The current review provides an overview of clear cell renal cancer. The authors valuablely address the epidemiology, diagnosis, and treatment. It sounds more like a commentary than a systematic or narrative review. It should be improved with a "Materials and Methods" paragraph that may highlight the search keyword and the number of papers on the field. The paper lacks several items, such as:

  1. Comment - It should be improved with a "Materials and Methods" paragraph that may highlight the search keyword and the number of papers on the field.
    1. Response – Thank you very much for your detailed review of this paper as it was extremely valuable for its final production. We appreciate your recommendation of adding materials and methods, however, our paper is a narrative review and most journals including the journal MPDI Cancers do not have a section for “Materials and Methods” for review papers. Hope this will address your concerns.
  2. Comment - “Preoperative scores predicting complications within the Partial nephrectomy part (PMID 39408076, 39369793, 34528773, 29715137) . PADUA, RENAL, SAFE as well as SPARE scores are commonly used in renal practice and should be added in a separate paragraph:
    1. Response – Thanks again. We agreed that the paper would benefit by adding scoring predictions of complications. We have used the papers you suggested and added multiple paragraphs to look at how different scores predicted outcomes for nephrectomies in our revised version.

  1. Comment - "Future Perspective" paragraph
    1. Response – Thanks again. This is a very constructive comment and we really appreciate it. As suggested, we have added a future perspectives paragraph that details the modern advances in adjuvant immunotherapy post-nephrectomy. The optimization of adjuvant therapy is a growing area in the field currently, that we aimed to validate with recent research. Further, we added a small section on novel research in the use of miRNAs to track disease progression.

  1. Comment – “papers on positive margin status and further treatments after PN (PMID 25450033, 29715137), sarcomatoid variants, the use of hemostatic agents (PMID 25139104)”
    1. Response - Thanks again. We added a future perspectives section which includes the use of further treatments after PN, such as VEGFR tyrosine kinase inhibitors, immune checkpoint inhibitors, and the use of miRNA as a biomarker to track disease progression. Though we did not include any section on the sarcomatoid variant of clear cell renal cell carcinoma, we did include a small section on the hemostatic agents study you mentioned, as it added an interesting finding in the area of partial nephrectomy.

We thank the editor and the reviewers for the additional opportunity to make changes/clarifications in this manuscript. We hope that these changes/clarifications will allow our manuscript to be accepted for publication in the prestigious journal MDPI: Cancers. 

Please do not hesitate to contact us if you need further information.  Again, thank you for reviewing our work. 

Respectfully submitted,

Yujiang Fang MD PhD

Reviewer 2 Report

Comments and Suggestions for Authors

The current paper aims to synthesize the epidemiology, the diagnosis, and the treatment of clear cell renal cancer. 

Despite the efforts, the current manuscript is only a paper list, confirming the EAU guidelines. First of all, a PRISMA statement should be added. Second, novel papers should be discussed. For example, the role of surgery in patients with a history of heart valve replacement (PMID 38526833), nephrometric score and surgical device (PMID 39408076, 39369793, 25139104 ), race ethnicity (PMID 33610487), as well as the histological variants (PMID 21871053, 32050629, 36906483). 

It should be improved in the discussion. A particular role should be acknowledged for the immunotherapy.  After those modifications, the paper might be suitable for any further consideration

Author Response

Thank you for reviewing our above-named manuscript submitted to MPDI Cancers.  We have made changes requested by the reviewers in our revision or addressed the concerns in this response, as detailed below and/or shown in the revised manuscript.

Reviewer #2: The current paper aims to synthesize the epidemiology, the diagnosis, and the treatment of clear cell renal cancer. Despite the efforts, the current manuscript is only a paper list, confirming the EAU guidelines. 

  1. Comment – “a PRISMA statement should be added”
    1. Response – Thank you very much for all of your remarks as they were extremely valuable. Because this is a narrative review and not a statistical meta-analysis, a PRISMA statement is not warranted as the scope of paper focuses broadly on multiple topics that are not numerically quantified.

  1. Comment – “novel papers should be discussed. For example, the role of surgery in patients with a history of heart valve replacement (PMID 38526833), nephrometric score and surgical device (PMID 39408076, 39369793, 25139104 ), race ethnicity (PMID 33610487), as well as the histological variants (PMID 21871053, 32050629, 36906483). “
    1. Response – Thanks again. Many of the papers you mentioned were extremely relevant for the partial nephrectomy section of the paper and were added. They added more modern insight, thus elevating the value and novelty of this paper. We did not add anything on various other histological variants, as we are attempting to keep this paper as focused as possible on the clear cell renal cell carcinoma variant. The recent publication on race/ethnicity determinants of surgical treatment was a great addition.

  1. Comment – “It should be improved in the discussion. A particular role should be acknowledged for the immunotherapy”
    1. Response - Thanks again. We commented on the use of ICIs becoming a neoadjuvant standard of care in cases of those affected by advanced clear cell renal cell carcinoma. Further we highlighted their future protentional in the use of adjuvant settings in the new Future Perspectives section that is now added, as there is a large increase in the body of work supporting their use in this setting.

We thank the editor and the reviewers for the additional opportunity to make changes/clarifications in this manuscript. We hope that these changes/clarifications will allow our manuscript to be accepted for publication in the prestigious journal MDPI: Cancers. 

Please do not hesitate to contact us if you need further information.  Again, thank you for reviewing our work. 

Respectfully submitted,

Yujiang Fang MD PhD

Reviewer 3 Report

Comments and Suggestions for Authors

Clear cell renal cell carcinoma (ccRCC) is the most common kidney cancer and is associated with poor outcomes, influenced by risk factors like smoking, obesity, hypertension, and genetic conditions such as tuberous sclerosis complex and Von Hippel-Lindau syndrome. Despite advances, current knowledge is outdated, with outdated diagnostic criteria and limited treatment options, relying mostly on surgery, though newer treatments targeting tumor growth pathways and immune checkpoints are emerging.

Major comments

1. This paper is a limited contribution to the field because it mostly reiterates established information about ccRCCs and does not offer much in the way of substantive new insights or perspectives.

2. Few papers within the past 5 years are cited, making it difficult to understand the current status of recent treatment for ccRCC.

3. Tyrosine kinase inhibitors such as Axitinib, Cabozantinib, Pazopanib, and Sorafenib could be discussed.

Minor comments

1. Sunitib should be corrected to Sunitinib.

2. Figure 1 and 2 have poor resolution and should be improved.

3. Write the full spellings of all abbreviations used in the figure in the figure legend.

Figure 1; CT, MRI, FDG-PET, CRP, LDH, CBC, ccRCC.

Figure 2; RTK, PD-1, VEGF

4. Reference 85: Add the year of issue.

Author Response

Thank you for reviewing our above-named manuscript submitted to MPDI Cancers.  We have made changes requested by the reviewers in our revision or addressed the concerns in this response, as detailed below and/or shown in the revised manuscript.

Reviewer #3: Clear cell renal cell carcinoma (ccRCC) is the most common kidney cancer and is associated with poor outcomes, influenced by risk factors like smoking, obesity, hypertension, and genetic conditions such as tuberous sclerosis complex and Von Hippel-Lindau syndrome. Despite advances, current knowledge is outdated, with outdated diagnostic criteria and limited treatment options, relying mostly on surgery, though newer treatments targeting tumor growth pathways and immune checkpoints are emerging.

  1. Comment – “This paper is a limited contribution to the field because it mostly reiterates established information about ccRCCs and does not offer much in the way of substantive new insights or perspectives.”
    1. Response – Thank you very much for your detailed review of this paper. It’s apparent that you reviewed it in close detail, which is extremely valuable in helping us produce the highest quality final version. We’ve added a future perspectives section that details the management of adjuvant therapy post-nephrectomy, which is a growing area in the field. All the papers mentioned in this new section were written in the past five years. Also, more modern references have been added throughout the entirety of the paper to ensure novelty as we want to ensure this adds value to the field.

  1. Comment – “Few papers within the past 5 years are cited, making it difficult to understand the current status of recent treatment for ccRCC.”
    1. Response – Thanks again. We’ve added a future perspectives section that details the management of adjuvant therapy post-nephrectomy, which is a growing area in the field. All the papers mentioned in this new section were written in the past five years. Also, more modern references have been added throughout this paper to ensure novelty.

  1. Comment – “Tyrosine kinase inhibitors such as Axitinib, Cabozantinib, Pazopanib, and Sorafenib could be discussed.”
    1. Response – Thanks again. We have incorporated studies including these drugs and their outcomes.

  1. Comment – “Sunitib should be corrected to Sunitinib”
    1. Response – Thanks again. We have corrected both spelling and capitalization in figure 2 per your mentioning this.

  1. Comment – “Figure 1 and 2 have poor resolution and should be improved.”
    1. Response – Thanks again. We will send in figure images to the editors office in the correct resolution format now.

  1. Comment – “Write the full spellings of all abbreviations used in the figure in the figure legend. Figure 1; CT, MRI, FDG-PET, CRP, LDH, CBC, ccRCC. Figure 2; RTK, PD-1, VEGF.”
    1. Response – Thanks again. We have corrected this under their corresponding figures.

  1. Comment – “Reference 85: Add the year of issue.”
    1. Response – Thanks again. We have corrected this citation in our revised version.

We thank the editor and the reviewers for the additional opportunity to make changes/clarifications in this manuscript. We hope that these changes/clarifications will allow our manuscript to be accepted for publication in the prestigious journal MDPI: Cancers. 

Please do not hesitate to contact us if you need further information.  Again, thank you for reviewing our work. 

Respectfully submitted,

Yujiang Fang MD PhD

Round 2

Reviewer 2 Report

Comments and Suggestions for Authors

The authors addressed properly my comments

Author Response

Thank you.

Reviewer 3 Report

Comments and Suggestions for Authors

The authors have answered all my questions and substantially revised the text so that it is acceptable to Cancers.

Author Response

Thank you.